# Comment on Pyrrolizidine Alkaloids and Terpenes from *Senecio* (Asteraceae): Chemistry and Research Gaps in Africa

**DOI:** 10.3390/molecules27248868

**Published:** 2022-12-13

**Authors:** Nicholas John Sadgrove

**Affiliations:** Department of Botany and Plant Biotechnology, University of Johannesburg (Auckland Park Campus), P.O. Box 524, Auckland Park, Johannesburg 2006, South Africa; nicholas.sadgrove@gmail.com

**Keywords:** pyrrolizidine, alkaloid, cacalol, eremophilane, contamination, adulteration, toxin, poison

## Abstract

The genus *Senecio* is one of the largest in Asteraceae. There are thousands of species across the globe, either confirmed or awaiting taxonomic delimitation. While the species are best known for the toxic pyrrolizidine alkaloids that contaminate honeys (as bees select pollen from the species) and teas via lateral transfer and accumulation from adjacent roots of *Senecio* in the rhizosphere, they are also associated with more serious cases leading to fatality of grazing ruminants or people by contamination or accidental harvesting for medicine. Surprisingly, there are significantly more sesquiterpenoid than pyrrolizidine alkaloid-containing species. The main chemical classes, aside from alkaloids, are flavonoids, cacalols, eremophilanes, and bisabolols, often in the form of furan derivatives or free acids. The chemistry of the species across the globe generally overlaps with the 469 confirmed species of Africa. A small number of species express multiple classes of compounds, meaning the presence of sesquiterpenes does not exclude alkaloids. It is possible that there are many species that express the pyrrolizidine alkaloids, in addition to the cacalols, eremophilanes, and bisabolols. The aim of the current communication is, thus, to identify the research gaps related to the chemistry of African species of *Senecio* and reveal the possible chemical groups in unexplored taxa by way of example, thereby creating a summary of references that could be used to guide chemical assignment in future studies.

## 1. Introduction

According to ‘Plants of the World Online’ (https://powo.science.kew.org/, accessed on 15 October 2022) (POWO), there are 1477 accepted species of *Senecio* in the world, with a further 3490 tentative species or synonyms, making it one of the largest genera in Asteraceae. According to POWO, 477 of the accepted species are native to Africa, but if the recognized synonyms are excluded [1,2,3], the total number of recognized African *Senecio* is reduced to 469, many of which are reported as a traditional medicine or a dangerous contaminant of foods and medicines [4,5,6].

It is the pyrrolizidine alkaloids in several species of *Senecio* that underly the poisonings occurring in ruminants [7] and people, leading to hepatomegaly (enlarged liver), ascites (abdominal fluid), and cirrhosis [4]. The toxicosis of people has been noted in cases of accidental contamination of a medicinal species [8], contamination of honey by bees [9], or by lateral transfer of toxic alkaloids in tea plantations [6], and water contamination [10,11,12].

In South Africa, *Senecio angustifolius* (Thunb.) Willd., contaminates Rooibos tea (*Aspalathus linearis* (Burm.f.) R.Dahlgren). Unfortunately, *S. angustifolius* has a similar growth habit and flower color as *A. linearis*, making it difficult to eliminate from Rooibos plantations. As the invading species grows among the Rooibos plants, it secretes pyrrolizidine alkaloids into the rhizosphere, where they are enter the root system of *A. linearis* and accumulate in the tea leaves [6].

Another example from South Africa is related to therapeutic use of *Senecio coronatus* (Thunb.) Harv. The roots of the species are used in traditional medicine. A suppository of the aqueous extract is given to infants as a means to confer strength to the child during weaning. Unfortunately, several infants have succumbed to hepatic sinusoidal obstruction syndrome, which is known to have been occurring since the 1980s [13], and possibly much longer. Due to superstition around the declaration of materials to forensics, it was not until 2017, following a new wave of deaths, that an examination of the biota could be made, revealing that the material associated with poisoning contained biota that was morphologically different, possibly representative of another species as a contaminant [8] or a toxic genotype currently incorrectly circumscribed as *S. coronatus* [14].

There are several species of *Senecio* that are associated with the toxication of honey. The presences of pyrrolizidine alkaloids in honeys has been reported in Europe [15], Brazil [9], China [16], North America [17], and via the South African species *S. inaequidens* DC [18], which is now naturalized in Italy (and Europe). The honey is toxified by the pollen from species of *Senecio*, harvested by bees, and carried back to the hive. The issue in Europe has prompted the European Food and Safety Authority to elaborate on the health risks associated with the consumption of honeys or plant products known to express or be contaminated by pyrrolizidine alkaloids [19]. The risks are also evident in Australia, since *Echium plantagineum* L., known by the vernacular ‘Patterson’s Curse’, expresses pyrrolizidine alkaloids that also find their way into honey. While the local honey, known as ‘Patterson’s Curse Honey’ is popular, it is contaminated with pyrrolizidine alkaloids [20].

Although the chemistry of *Senecio* is well-known to include pyrrolizidine alkaloids, and the genus is chemically varied. Surprisingly, there are many more terpenoid taxa than toxic species in *Senecio*. Furthermore, there are several chemical studies of species that were previously circumscribed as *Senecio*, but they are now revised to such genera as *Caputia* or *Othonna* [1] (among others). While the taxonomy has changed, the chemical similarities to species in *Senecio* are evident, which can be ascertained by reading the previous papers by Bohlmann [21,22]. In another example, the macrolide platyphylline and the pyrrolizidine alkaloid seneciphylline were first discovered after isolation from *Senecio platyphyllus* D.C. [1]. The etymology of the vernacular names given to these alkaloids is related to the botanical name of the species. However, *S. platyphyllus* was renamed to *Caucasalia macrophylla* (M.Bieb.) B.Nord. There are many examples of species revisions that complicate the interpretation of the etymology of the compound names.

The current communication is a summary of the chemistry and toxicity of African species of *Senecio*. This work is intended to serve as a reference in guiding the further chemical prospection of the species in southern Africa. The correct names of all taxa were determined using the POWO database to remain up to date with the taxonomic status of all species listed. A literature search was conducted on each species individually to ascertain if the chemical characterization was retrievable, to identify what is known, the chemotaxonomic implications thus far, and to reveal research gaps. The literature search was also extended to species synonyms whenever old or outdated names were realized in the course of compiling data.

## 2. Phytochemistry of African *Senecio*

A search of the 469 species of *Senecio* from Africa was conducted on Google Scholar [23] to ascertain if phytochemical studies exist, individually searching the genus and species against the words ‘chemistry’, ‘phytochemistry’, ‘sesquiterpene’, or ‘pyrrolizidine alkaloid’. The species (or subspecies) searched are listed in Table 1, and the tentative number of species associated with phytochemical studies amounted to 83 (Table 2).

The species with phytochemical records, listed in Table 1, have been elaborated upon in Table 2, with details related to the chemical classes identified.

The most common classes of compound identified and published from African *Senecio* include pyrrolizidine alkaloids, eremophilanes, biasabolols, cacalols, and flavonoids (Figure 1). Derivatives of these include furans, oxides, O-linked moieties, and mere stereoisomers. Other groups of compounds that are less frequently reported include phenylpropanes, essential oils, triterpenes, sterols, oxyeuryopsins, diterpenes, dimers of sesquiterpenes, fatty acid derivatives, and polyunsaturated alkynes and alkenes (Table 2).

The vast majority of the chemical work performed on the world’s species of *Senecio* was conducted in Germany by Bohlmann and collaborators [21,22,33,41,43,45,55,58,59,61,72,75,79,81,82,84,85,90,94,98]. This group covered a number of the African species, sometimes focused exclusively on those from South Africa (südafrikanischen) [21,22,90]. The majority of the work by Bolmann and his group characterized a significant number of new and existing sesquiterpenes, particularly the cacalols and their derivatives. The cacalols were named from the species *Cacalia delphiniifolia* Siebold and Zucc. [105], and although the species was revised to *Japonicalia delphiniifolia* (Siebold and Zucc.) C. Ren and Q.E. Yang, the cacalols are now widely characterized in the genus *Senecio* (Table 2).

The contents of Table 2 are not representative of exhaustive chemical studies of the respective species. For example, where essential oils have been tentatively characterized, the fixed components of the material have not been studied. Furthermore, in a minority of cases, multiple types of chemical classes are identified, sometimes within a single study. For example, *S. vira vira* was exhaustively studied, and flavonoids, sterols, triterpenes, and pyrrolizidine alkaloids were identified in that single biota. This underscores the possibility of other classes of metabolite in the other species in Table 2. When chemists study pyrrolizidine alkaloids, they focus on an ‘alkaloids’ extract produced via acid-base partitioning [106]. By following such an approach, non-basic metabolites are excluded from the extracts. Thus, the species listed in Table 2 may be more complex than is realized.

A chemotaxonomic review of *Senecio* divides the genus into sections, corresponding to the different types of sesquiterpenes, i.e., section bisabolene, sect. cacalol, sect. eremophilane, sect. furanoeremophilane, sect. eremophilanolide, and sect. germacrane [29]. The approach to recognize taxa, according to a specific group of metabolites (the sesquiterpenes), to the exclusion of other metabolites that may be present in the biota, reduces the complexity of the analysis. This approach may have been informed by the impractical division of alkaloids verses terpenes, due to the technical difficulty mentioned above. However, to know if the clustering tree of sesquiterpenes in *Senecio* is of true value, it may be necessary to test for agreement of a molecular phylogeny [107].

## 3. Conclusions

The genus *Senecio* is one of the largest in the family Asteraceae. Species in the genus are chemically varied, and previous scholars have attempted to understand this from a chemotaxonomic perspective. Several of the species have been associated with poisonings of both ruminants and humans, ranging from mild toxicosis to fatality. The pyrrolizidine alkaloids are the toxic principle of the poisonous species, and while there are many species that are not reported to contain this class of compound, studies do not always focus on this class, nor do they confirm that they searched for them in extracts or qualify that they are not present. Thus, the chemical information that has been gathered in this communication is not a robust guide to the exhaustive chemistry of the respective species, but rather a preliminary finding that can be elaborated upon with further study. Thus, it is of essence to re-examine the species that reportedly contain sesquiterpenes to know if pyrrolizidine alkaloids are absent generally, or if the co-occurrence of the two classes of compound is common.

## Figures and Tables

**Figure 1 molecules-27-08868-f001:**
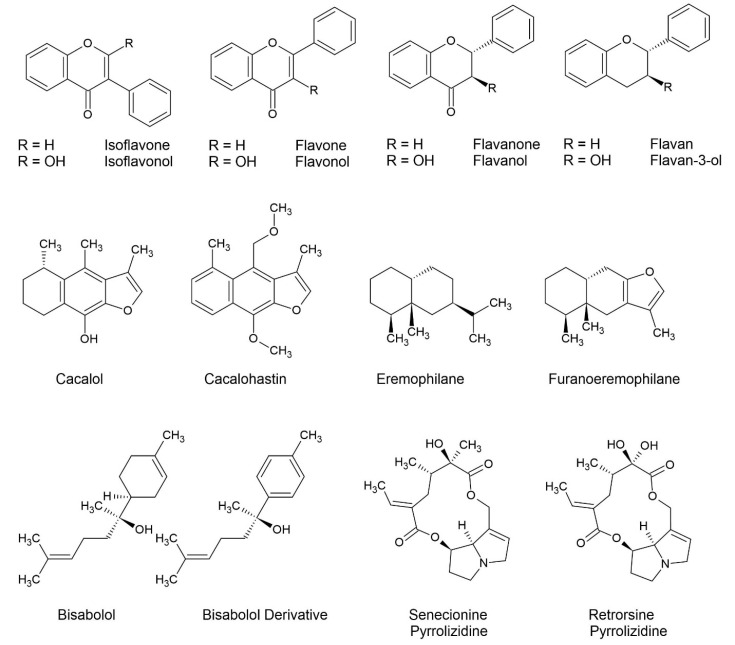
Common classes of compound in African species of *Senecio*.

**Table 1 molecules-27-08868-t001:** A list of all of the currently accepted species of *Senecio* that are native to Africa, according to POWO (477 species, 469 after synonyms are subtracted), conveying those with phytochemical information and those with no records listed here, as they were not identified in the literature search. Y = phytochemical study found, N = no phytochemical study found.

Species and Author	Y/N	Species and Author	Y/N	Species and Author	Y/N
*S. abbreviatus* S.Moore	N	*S. glutinosus* Thunb.	Y	*S. paniculatus* P.J. Bergius	Y
*S. abruptus* Thunb.	N	*S. gossweileri* Torre	N	*S. parascitus* Hilliard	N
*S. acetosifolius* Baker	N	*S. gossypinus* Baker	Y	*S. parentalis* Hilliard and B.L. Burtt	N
*S. achilleifolius* DC.	N	*S. gramineticola* C. Jeffrey	N	*S. parvifolius* DC. (Synonym of *S. carroensis*)	N
*S. actinoleucus* F.Muell.	N	*S. gramineus* Harv.	N	*S. paucicalyculatus* Klatt	Y
*S. acutifolius* DC.	N	*S. grandiflorus* P.J. Bergius	Y	*S. pauciflorus* Baker	N
*S. adenostylifolius* Humbert	N	*S. gregatus* Hilliard	N	*S. pauciflosculosus* C. Jeffrey	N
*S. adnatus* DC.	Y	*S. hadiensis* Forssk.	Y	*S. pearsonii* Hutch. (Synonym of *S. asperulus*)	N
*S. adscendens* (*syn. andinus*) Bojer	N	*S. halimifolius* L.	Y	*S. pellucidus* DC.	N
*S. aegyptius* L.	Y	*S. harveyanus* MacOwan	N	*S. peltophorus* Brenan	N
*S. aequinoctialis* R.E.Fr.	N	*S. hastatus* L. (Synonym of *S. erosus* and *S. robertiifolius*)	N	*S. penninervius* DC.	N
*S. aetfatensis* B.Nord.	N	*S. hastifolius* (L.f.) Less.	N	*S. pentactinus* Klatt	N
*S. affinis* DC.	Y	*S. haygarthii* *M.Taylor* ex Hilliard	N	*S. pentecostus* Hiern	N
*S. agapetes* C.Jeffrey	N	*S. hebdingii* (Rauh and Buchloh) G.D. Rowley	N	*S. perralderianus* Coss.	N
*S. albanensis* DC.	N	*S. hedbergii* C. Jeffrey	N	*S. perrieri* Humbert	N
*S. albanopsis* Hilliard	N	*S. hederiformis* Cron	N	*S. perrottetii* DC.	N
*S. albifolius* DC.	N	*S. heliopsis* Hilliard and B.L. Burtt	Y	*S. persicifolius* L.	N
*S. albopunctatus* Bolus	N	*S. helminthioides* (Sch.Bip.) Hilliard	Y	*S. petiolaris* DC.	N
*S. aloides* DC.	N	*S. hermannii* B. Nord.	N	*S. petraeus* Boiss. and Reut.	N
*S. altissimus* Mill.	N	*S. hesperidum* Jahand, Maire, and Weiller	N	*S. phalacrolaenus* DC.	N
*S. ambositrensis* Humbert	N	*S. hieracioides* DC.	Y	*S. pillansii* Levyns	N
*S. amplificatus* C.Jeffrey	N	*S. hildebrandtii* Baker	N	*S. pinifolius* (L.) Spreng.	N
*S. anapetes* C.Jeffrey	N	*S. hirsutilobus* Hilliard	N	*S. pinnatifidus* Less.	N
*S. andapensis* Humbert	N	*S. hirtifolius* DC	N	*S. pinnatipartitus* Sch.Bip. ex Oliv.	N
*S. andohahelensis* Humbert	N	*S. hirto-crassus* Humbert	N	*S. pinnulatus* Thunb.	N
*S. angulatus* L.f.	Y	*S. hochstetteri* Sch.Bip. Ex A.Rich.	N	*S. piptocoma* O.Hoffm.	N
*S. angustifolius* (Thunb.) Willd.	Y	*S. hoggariensis* Batt. and Trab.	Y	*S. pirottae* Chiov.	N
*S. anomalochrous* Hilliard	N	*S. hollandii* Compton	N	*S. plantagineoides* C. Jeffrey	N
*S. antaisaka* Humbert	N	*S. holubii* Hutch. and Burtt Davy	N	*S. pleistophyllus* C. Jeffrey	N
*S. antambolorum* Humbert	N	*S. humidanus* C. Jeffrey	N	*S. poggeanus* Mattf.	N
*S. antandroi* Scott Elliot	N	*S. hypochoerideus* DC.	Y	*S. polelensis* Hilliard	N
*S. anthemifolius* Harv.	N	*S. ilicifolius* Thunb.	Y	*S. polyadenus* Hedberg	N
*S. antitensis* Baker	N	*S. ilsae* A.Santos and ReY-Bet	N	*S. polyanthemoides* Sch.Bip.	Y
*S. aquifoliaceus* DC.	N	*S. immixtus* C. Jeffrey	N	*S. polyodon* DC.	N
*S. arabidifolius* O.Hoffm	N	*S. inaequidens* DC.	Y	*S. poseideonis* Hilliard and B.L. Burtt	N
*S. arenarius* Thunb.	N	*S. incomptus* DC.	N	*S. praeteritus* Killick	N
*S. arniciflorus* DC	N	*S. incrassatus* Lowe	N	*S. propior* S. Moore	N
*S. asperulus* DC	Y	*S. infirmus* C. Jeffrey	N	*S. prostratus* Klatt	N
*S. auriculatissimus* Britton	N	*S. ingeliensis* Hilliard	N	*S. pseudolongifolius* Sch.Bip ex J. Calvo	N
*S. austromontanus* Hilliard	N	*S. inornatus* DC.	Y	*S. pseudosubsessilis* C. Jeffrey	N
*S. balensis* S.Oritz and Vivero	N	*S. intricatus* S.Moore	N	*S. ptarmicifolius Bory*	N
*S. bampsianus* Lisowski	N	*S. isatideus* DC.	Y	*S. pterophorus* DC.	Y
*S. barbatus* DC.	N	*S. isatidoides* E. Phillips and C.A.Sm.	N	*S. puberulus* DC.	N
*S. baronii* Humbert	N	*S. jacksonii* S. Moore	N	*S. pubigerus* L.	Y
*S. barorum* Humbert	N	*S. junceus* (Less.) Harv.	N	*S. purpureus* L.	Y
*S. basalticus* Hilliard	N	*S. juniperinus* L.f.	N	*S. purtschelleri* Engl.	N
*S. baurii* Oliv.	N	*S. junodii* Hutch. and Burtt Davy	N	*S. qathlambanus* Hilliard	N
*S. belbeysius* Delile	N	*S. kacondensis* S. Moore	N	*S. quartziticola* Humbert	N
*S. bellis* Harv.	N	*S. kalambatitrensis* Humbert	N	*S. quinquelobus* (Thunb.) DC.	N
*S. bipinnatus* Less.	N	*S. kalingenwae* Hilliard and B.L. Burtt	N	*S. quinquenervius* Sond.	N
*S.* Bolle*i Sunding* and *C.Kunkel*	Y	*S. karaguensis* O. Hoffm.	N	*S. ragazzii* Chiov.	N
*S. boutonii* Baker	N	*S. katangensis* O. Hoffm.	N	*S. randii* S.Moore	N
*S. brachyantherus* (Hiern) S.Moore	N	*S. kayomborum* Beentje	N	*S. rehmannii* Bolus	N
*S. brachypodus* DC	Y	*S. keniophytum* R.E.Fr.	N	*S. repandus* Thunb.	N
*S. brevidentatus* M.D. Hend	N	*S. kerdousianus* Gomiz and Llamas	N	*S. reptans* Turcz.	N
*S. brevilorus* Hilliard	N	*S. kuluensis* S.Moore	N	*S. resectus* Bojer ex DC.	N
*S. brittenianus* Hiern	N	*S. kundelungensis* Lisowski	N	*S. retortus* Benth.	N
*S. bryoniifolius* Harv.	N	*S. kuntzeanus* Dinter	N	*S. retrorsus* DC.	Y
*S. bulbinefolius* DC.	N	*S. laevigatus* Thunb.	N	*S. rhammatophyllus* Matt.f.	N
*S. bupleuroides* DC.	Y	*S. laevis* Humbert	N	*S. rhomboideus* Harv.	Y
*S. burchellii* DC.	Y	*S. lamarckianus* Bullock	N	*S. rhyncholaenus* DC.	N
*S. burtonii* Hook.f.	Y	*S. lanceus* Aiton	N	*S. rigidus* L.	N
*S. byrnensis* Hilliard	N	*S. latecorymbosus* Gilli	N	*S. robertiifolius* DC.	N
*S. cakilefolius* DC. (Synonym of *S. arenarius*)	N	*S. latibracteatus* Humbert	N	*S. roseiflorus* R.E.Fr.	Y
*S. caloneotes* Hilliard	N	*S. laticipes* Bruyns	N	*S. rosmarinifolius* L.f.	Y
*S. canabyi* Humbert	N	*S. latifolius* DC.	Y	*S. rugegensis* Muschl.	N
*S. canaliculatus* Bojer ex DC.	N	*S. latissimifolius* S. Moore	N	*S. ruwenzoriensis* S. Moore	Y
*S. canalipes* DC.	N	*S. lawalreeanus* Lisowski	N	*S. sabinjoensis* Muschl.	N
*S. capuronii* Humbert	N	*S. laxus* DC.	N	*S. saboureaui* Humbert	N
*S. cardaminifolius* DC.	N	*S. leandrii* Humbert	N	*S. sakalavorum* Humbert	N
*S. carnosus* Thunb.	N	*S. lejolyanus* Lisowski	N	*S. sakamaliensis* (Humbert) Humbert	N
*S. carroensis* DC.	N	*S. lelyi* Hutch.	N	*S. sandersonii* Harv.	Y
*S. cathcartensis* O. Hoffm.	Y	*S. leptopterus* Mesfin	N	*S. saniensis* Hilliard and B.L. Burtt	N
*S. caudatus* DC.	N	*S. lessingii* Harv.	N	*S. schimperi* Sch.Bip. ex Hochst.	N
*S. cedrorum* Raynal	N	*S. letouzeyanus* Lisowski	N	*S. schultzii* Hochst. ex A. Rich.	N
*S. chalureaui* Humbert	N	*S. leucadendron* (G. Forst.) Hems.L.	N	*S. schweinfurthii* O.Hoffm.	Y
*S. chrysocoma* Meerb.	Y	*S. leucanthemifolius Poir.*	Y	*S. scitus* Hutch. and Burtt Davy	N
*S. cinerascens* Aiton	N	*S. lewallei* Lisowski	N	*S. scoparius* Harv.	N
*S. citriceps* Hilliard and B.L.Burtt	N	*S. lineatus* DC.	N	*S. semiampl*ex*ifolius* De Wild.	N
*S. cochlearifolius* Bojer ex DC.	N	*S. linifolius* L.	Y	*S. seminiveus* J.M. Wood and M.S. Evans	N
*S. coleophyllus* Turcz.	N	*S.* Lisowski*i* Long Wang and Beentje	N	*S. serratuloides* DC.	Y
*S. comptonii* J.C.Manning and Goldblatt	N	*S. litorosus* Fourc.	N	*S. serrulatus* DC.	N
*S. confertus* Sch.Bip. Ex A.Rich.	N	*S. littoreus* Thunb.	N	*S. serrurioides* Turcz.	N
*S. conrathii* N.E.Br.	Y	*S. lividus* L.	Y	*S. shabensis* Lisowski	N
*S. consanguineus* DC.	Y	*S. lobelioides* DC. (Synonym of *S. flavus*)	N	*S. simplicissimus* Bojer ex DC.	N
*S. cordifolius* L.f.	N	*S. longiscapus* Bojer ex DC.	Y	*S. sisymbriifolius* DC.	N
*S. cornu-cervi* MacOwan	N	*S. luembensis* De Wild. and Muschl.	N	*S. skirrhodon* DC.	N
*S. coronatus* (Thunb.) Harv.	Y	*S. lycopodioides* Schltr.	N	*S. snowdenii* Hutch.	N
*S. cotyledonis* DC.	N	*S. lydenburgensis* Hutch. and Burtt Davy	Y	*S. sociorum* Bolus	N
*S. crassissimus* Humbert	Y	*S. lygodes* Hiern	N	*S. sophioides* DC.	N
*S. crassiusculus* DC.	N	*S. lyratus* Forssk.	Y	*S. sororius* C. Jeffrey	N
*S. crassorhizus* De	N	*S. mabberleyi* C.Jeffrey	N	*S. sotikensis* S. Moore	N
*S. crenatus* Thunb.	N	*S.* MacOwan*ii* Hilliard	N	*S. spartareus* S. Moore	N
*S. crenulatus* DC.	N	*S. macrocephalus* DC.	Y	*S. speciosissimus* J.C.Manning and Goldblatt	N
*S. crispatipilosus* C.Jeffrey	N	*S. macroglossoides* Hilliard	N	*S. speciosus* Willd.	Y
*S. crispus* Thunb.	Y	*S. macroglossus* DC.	N	*S. spiraeifolius* Thunb.	N
*S. cristimontanus* Hilliard	N	*S. macrospermus* DC.	Y	*S. squalidus* L.	Y
*S. cryphiactis* O.Hoffm.	N	*S. madagascariensis Poir.*	Y	*S. stella-purpurea* V.R.Clark, J.D.Vidal, and N.P.Barker	N
*S. cryptolanatus* Killick	N	*S. malacitanus Huter*	Y	*S. steudelii* Sch.Bip. ex A. Rich.	N
*S. cyaneus* O.Hoffm.	N	*S. malaissei* Lisowski	N	*S. striatifolius* DC.	N
*S. cymbalariifolius* (L.) Less.	N	*S. mandrarensis* Humbert	N	*S. strictifolius* Hiern	N
*S. decaryi* Humbert	N	*S. maranguensis* O.Hoffm.	N	*S. subcanescens* (DC.) Compton	N
*S. decurrens* DC.	N	*S. margaritae* C.Jeffrey	N	*S. subcoriaceus* Schltr.	N
*S. deltoideus* Less.	Y	*S. marginalis* Hilliard	N	*S. subfractifl*ex*us* C. Jeffrey	N
*S. denisii* Humbert	N	*S. mariettae* Muschl.	Y	*S. submontanus* Hilliard and B.L. Burtt	N
*S. dentatoalatus* Mildbr. Ex C.Jeffrey	N	*S. maritimus* L.f.	Y	*S. subrubriflorus* O.Hoffm.	Y
*S. depauperatus* Mattf.	N	*S. marnieri* Humbert	N	*S. subsessilis* Oliv. and Hiern	N
*S. diffusus* L.f.	N	*S. marojejyensis* Humbert	N	*S. subsinuatus* DC.	N
*S. digitalifolius* DC.	N	*S. massaicus* (Maire) Maire	Y	*S. sylvaticus* L.	Y
*S. dilungensis* Lisowski	N	*S. matricariifolius* DC.	N	*S. syringifolius* O.Hoffm.	Y
*S. diodon* DC.	N	*S. mattirolii* Chiov.	N	*S. tabulicola* Baker	N
*S. diphyllus* De Wild. and Muschl.	N	*S. mauricei* Hilliard and B.L. Burtt	Y	*S. tamoides* DC.	Y
*S. discodregeanus* Hilliard and B.L.Burtt	N	*S. maydae* Merxm. (Synonym of *S. albopunctatus*)	N	*S. tanacetopsis* Hilliard	N
*S. discokaraguensis* C.Jeffrey	N	*S. mbuluzensis* Compton	N	*S. teixeirae* Torre	N
*S. dissidens* Fourc.	N	*S. melastomifolius* Baker	N	*S. telekii* O.Hoffm.	N
*S. dissimulans* Hilliard and B.L.Burtt	N	*S. mesembryanthemoides* Bojer ex DC.	N	*S. telmateius* Hilliard	N
*S. doryphoroides* C.Jeffrey	N	*S. mesogrammoides* O.Hoffm.	N	*S. tenellus* DC.	N
*S. doryphorus* Mattf.	N	*S. meuselii* Rauh	N	*S. teneriffae* Sch.Bip. ex Bolle	N
*S. dracunculoides* DC.	N	*S. meyeri-johannis* EngL.	N	*S. thamathuensis* Hilliard	N
*S. dregeanus* DC.	N	*S. microalatus* C. Jeffrey	N	*S. thunbergii* Harv.	N
*S. dumeticola* S.Moore	N	*S. microglossus* DC.	Y	*S. torticaulis* Merxm.	N
*S. dumosus* Fourc.	N	*S. microspermus* DC.	N	*S. tortuosus* DC.	N
*S. eenii* (S.Moore) Merxm.	N	*S. mimetes* Hutch. and R.A. Dyer	N	*S. trachylaenus* Harv.	N
*S. elegans* L.	N	*S. mitophyllus* C. Jeffrey	N	*S. trachyphyllus* Schltr.	N
*S. ellenbeckii* O.Hoffm.	N	*S. mlilwanensis* Compton	N	*S. transmarinus* S. Moore	N
*S. eminens* Compton	N	*S. monticola* DC.	N	*S. triactinus* S. Moore	N
*S. emirnensis* DC.	N	*S. mooreanus* Hutch. and Burtt Davy	N	*S. trilobus* L.	N
*S. englerianus* O.Hoffm.	N	*S. moorei* R.E.Fr.	N	*S. triodontiphyllus* C. Jeffrey	N
*S. eriobasis* DC. (Synonym of *S. erosus*)	N	*S. mooreioides* C. Jeffrey	N	*S. triplinervius* DC.	N
*S. eriopus* Willk.	N	*S. morotonensis* C. Jeffrey	N	*S. triqueter* Less.	N
*S. erlangeri* O.Hoffm.	N	*S. mu*Cron*atus* Willd.	N	*S. tsaratananensis* Humbert	N
*S. erosus* L.f.	N	*S. multibracteatus* Harv.	N	*S. tugelensis* J.M. Wood and M.S. Evans	N
*S. erubescens* Aiton	Y	*S. multicaulis* DC.	N	*S. tysonii* MacOwan	N
*S. erysimoides* DC.	N	*S. multidenticulatus* Humbert	N	*S. ulopterus* Thell.	N
*S. esterhuyseniae* J.C.Manning and Goldblatt	N	*S. muricatus* Thunb.	N	*S. umbellatus* L.	Y
*S. euriopoides* DC.	N	*S. myriocephalus* Sch.Bip. ex A.Rich.	N	*S. umbricola* Cron and B. Nord.	N
*S. evelynae* Muschl.	N	*S. nanus* Sch.Bip. ex A. Rich.	N	*S. umgeniensis* Thell.	N
*S.* ex*arachnoideus* C.Jeffrey	N	*S. napifolius* MacOwan	N	*S. unionis* Sch.Bip. ex A. Rich.	N
*S.* ex*uberans* R.A.Dyer	N	*S. natalicola* Hilliard	N	*S. urophyllus* Conrath	N
*S. fanshawei* Beentje	N	*S. navicularis* Humbert	N	*S. urundensis* S.Moore	N
*S. farinaceus* Sch.Bip. Ex A.Rich.	N	*S. navugabensis* C. Jeffrey	N	*S. vaingaindrani* Scott Elliot	N
*S. flavus* (Decne.) Sch.Bip.	Y	*S. neo* Baker*i* Humbert	N	*S. variabilis* Sch.Bip.	N
*S. foeniculoides* Harv.	N	*S. neoviscidulus* Soldano	N	*S. venosus* Harv.	N
*S. forbesii* Oliv. and Hiern	N	*S. ngandae* Beentje	N	*S. verbascifolius* Burm.f.	N
*S. francoisii* Humbert	N	*S. ngoyanus* Hilliard	N	*S. vernalis* Walds. and Kit.	Y
*S. fresenii* Sch.Bip.	N	*S. nyangani* Beentje	N	*S. vestitus* P.J.Bergius	N
*S. gallicus* Vill. Ex Chaix	Y	*S. nyungwensis* P. Maquet	N	*S. vicinus* S.Moore	N
*S. gariepiensis* Cron	N	*S. ochrocarpus* Oliv. and Hiern	N	*S. villifructus* Hilliard	N
*S. garnieri* Klatt	N	*S. odontopterus* DC.	N	*S. vimineus* DC.	N
*S. gazensis* S.Moore	N	*S. oederifolius* DC.	N	*S. vira-vira* Hieron.	Y
*S. geniorum* Humbert	N	*S. ornatus* S.Moore	N	*S. vitellinoides* Merxm.	N
*S. gerrardii* Harv.	Y	*S. othonniflorus* DC.	N	*S. vittarifolius* Bojer ex DC.	N
*S. giessii* Merxm.	N	*S. oxyodontus* DC.	Y	*S. voigtii* van Jaarzv.	N
*S. glaberrimus* DC.	N	*S. oxyriifolius* DC.	Y	*S. volcanicola* C. Jeffrey	N
*S. glanduloso-lanosus* Thell.	N	*S. paarlensis* DC.	N	*S. vulgaris* L.	Y
*S. glandulo-pilosus* Volkens and Muschl.	N	*S. pachyrhizus* O. Hoffm.	N	*S. waterbergensis* S. Moore	N
*S. glastifolius* L.f.	N	*S. paludaffinis* Hilliard	Y	*S. windhoekensis* Merxm.	N
*S. glaucus* L.	Y	*S. panduratus* Less. (Synonym of *S. erosus*)	N	*S. wittebergensis* Compton	N
*S. glutinarius* DC.	N	*S. panduriformis* Hilliard	N	*S. xenostylus* O. Hoffm.	N

**Table 2 molecules-27-08868-t002:** Details of phytochemical studies of the 83 species (or subspecies) of *Senecio* in Africa, for which records could be found.

Species	Details
*S. adnatus* DC.	Toxic alkaloid (macrolide): Platyphylline [24].
*S. aegyptius* L.	Toxic alkaloids: Senecionine (pyrrolizidine) and otosenine (macrolide) [25].
*S. aegyptius* L. var. *discoideus* Boiss.	Sesquiterpenes: 1,10-epoxyfuranoeremophilane (in essential oil), with traces of monoterpenes [26]. Non-volatiles include 1-β-hydroxy-8-oxoeremophila-7,9-dien-12-oic acid, rutin, and quercetin-3-O-glucoside-7-O-rutinoside [27]. Novel eremophilane lactones also described [28].
*S. affinis* DC.	Sesquiterpenes: Cacalols and bisabolols [29].
*S. angulatus* L.f.	Toxic alkaloid: Angularine (pyrrolizidine) [30]. Phenols: Cynarin, chlorogenic acid and trans-ferulic acid [31]. Essential oils: α-Pinene, β-pinene, limonene, camphene, germacrene D, viridifloral, β-caryophyllene [32].
*S. angustifolius* (Thunb.) Willd.	Toxic alkaloids (pyrrolizidines): Senecionine N-oxide, retrorsine N-oxide, retrorsine, seneciphyline, senecionine, senkirkine [6].
*S. asperulus* DC.	Possible chemotypes. Terpenes: furoeremophilanes, α -humulene, ent-kaurenic acid, ent-kaurenol [33]. Toxic alkaloids: Pyrrolizidine alkaloid N-oxides (exact identity not known) [34].
*S. bollei* Sunding and C.Kunkel	Toxic alkaloid (pyrrolizidine): Senecivernine [35].
*S. brachypodus* DC.	Toxic alkaloid (pyrrolizidine): Rosmarinine [36].
*S. bupleuroides* DC.	Toxic alkaloid (pyrrolizidine): Retrorsine [37].
*S. burchellii* DC.	Toxic alkaloids (pyrrolizidine): Senecionine N-oxide and senkirkine [6].
*S. burtonii* Hook.F.	Sesquiterpene: Cacalolide derivative, 4α-[2′-hydroxymethylacryloxy]-1β-hydroxy-14-(5 → 6) abe oeremophilan-12,8-olide. Shikimic acid derivative, (3′E)-(1α)-3-hydroxymethyl-4β,5α-dimethoxycyclohex-2-enyloctadec-3′-enoate. Fatty acid derivatives, octacosan-1-ol, 3β-hydroxyolean-12-en-28-oic acid, and 3β-acetoxyolean-12-en-28-oic acid [38].
*S. cathcartensis* O.Hoffm.	Sesquiterpenes: Eremophilene derivatives [21].
*S. chrysocoma* Meerb.	Toxic alkaloids: 7-angelylplatynecine, 9-angelylplatynecine, sarracine, neosarracine [39], and other 7-Angelyl-1-methylenepyrrolizidines (pyrrolizidines) [40].
*S. conrathii* N.E.Br.	Sesquiterpenes: β-Farnesene, furoeremophilane derivatives, and germacrene D-4-ol [41]. Additionally, a nickel hyperaccumulator.
*S. consanguineous* DC.	Toxic alkaloid: Retrorsine (pyrrolizidine), very low concentration [42].
*S. coronatus* (Thunb.) Harv.	Species did not contain toxic alkaloids [8], or mere traces, but further work necessary to know of the chemistry.
*S. crassissimus* Humbert	Sesquiterpenes: Germacrene D, bicyclogermacrene, *Z*-caryophyllene epoxide and other epoxides. Triterpenes: Lupeol, its acetate, lupeone, β-amyrin acetate, β-amyrenone, glutin-5(6)-en-3β-ol, 28-oxo-β-amyrenone, and the angelate [43].
*S. crispus* Thunb.	Sesquiterpene dimers: Disesquiterpenoid derivative [44].
*S. deltoideus* Less.	Unusual sesquiterpenes, linear diterpenes, and polyunsaturated alkenes and -kynes [45]. Used as a medicine to treat gynaecological and obstetric disorders [46].
*S. erubescens* Aiton	Sesquiterpenes: Bisabolanes and eremophilanes [29].
*S. flavus* (Decne.) Sch.Bip.	Sesquiterpenes (Oxyeuryopsin derivatives): 3β-Methylbutyryloxyeuryopsin, 3β-angeloyloxyeuryopsin, 3β-senecioyloxyeuryopsin, 3β-hydroxyeuryopsin, euryopsin-3-one, furanoligularenone, and others [47].
*S. gallicus* Vill. Ex Chaix	Toxic alkaloids (pyrrolizidine): Ligularizine, senkirkine and senecionine N-oxide [48]. Essential oil: β-Phellandrene, apinene, germacrene-D, myrcene, α-copaene, sabinene, (*Z*)-β-ocimene, β-caryophyllene, *p*-cymene, β-pinene, α phellandrene, α-terpinolene, (*E*)-β-ocimene, α-humulene, azingiberene, and caryophyllene oxide [49].
*S. gerrardii* Harv.	Sesquiterpenes: Eremophilene derivatives [21].
*S. glaucus* L.	Essential oils: Isolongifolen-9-one, longiverbenone, 4-carene, p-cymene, thujone [50], m-mentha-1(7),8-diene, cis-m-mentha-2,8-diene, dehydrofukinone, α-terpinolene, 2,5-cyclohexadiene-1,4-dione,2-(1,1-dimethylethyl)-5-(2-methyl-2-propen-1-yl), sabinene, α-Fenchene and 1,3,8-p-menthatriene [51]. Phenols: Vanillic acid and gallic acid [52]. Flavonoids: Isorhamentin 3-O-β-D-glucoside, and isorhamentin 3-O-β-D-rutinoside. Benzofuran glucosides: 2,3-Dihydro-3β-hydroxyeuparin 3-O-glucopyranoside, isorhamentin 3-O-β-D-glucoside, and isorhamentin 3-O-β-D-rutinoside [53].
*S. glaucus* L. subsp. *coronopifolius* (syn. *S. desfontainei*).	Toxic alkaloids (pyrrolyzidine): Seneciphylline [25].
*S. glutinosus* Thunb.	Sesquiterpenes (seco-eremophilanes): Senglutinosin, 3α-hydroxy-10β-H-eremophil-11(13)-en-9-one, and nor-seco- glutinosone [33].
*S. gossypinus* Baker	Flavonoid: Kaempferol-3-O-α-L-arabinopyranoside. Triterpenes: α-Amyrin and β-amyrin [54].
*S. grandiflorus* P.J.Bergius	Sesquiterpenes (furano): Cacalol derivatives [55].
*S. hadiensis* Forssk.	Toxic alkaloids (macrolides): Rosmarinine, 12-O-acetylrosmarinine, neorosmarinine, hadiensine (1α-hydroxyplacyphylline), 12-O-acetylhadiewine, 12-O-acetylneohadiewine, and petitianine (2α-hydroxy-1,2-dihydroretronine) [56]. Sesquiterpenes (tricyclic): presilphiperfolan-2α,5α,8α-triol and presilphiperfolan-2α,5α,8α,10α-tetraol [57].
*S. halimifolius* L.	Sesquiterpenes (furano): Furanoeremophilanes [58].
*S. heliopsis* Hilliard and B.L.Burtt	Sesquiterpenes: Furanoeremophilanes and cacalols [29].
*S. helminthioides* (Sch.Bip.) Hilliard	Phenylpropenes: Trimethoxy-phenylpropenes [22].
*S. hieracioides* DC.	Sesquiterpenes (furo): 9,10-Dehydrofuranoeremophilane, ligularenolide, eremophil-7(11)-en-8,12-olide, 8,8′-epimeric dimers. Shikimic acid derivatives [59].
*S. hoggariensis* Batt. and Trab.	Phenols: 3,4-Dihydroxybenzoic acid, 4-hydroxybenzoic acid, 6,7-dihydroxycoumarin, vanillic acid, caffeic acid, *p*-coumaric acid, and ferulic acid [60].
*S. hypochoerideus* DC.	Sesquiterpenes (furano): Furanoeremophilanes [61].
*S. ilicifolius* Thunb.	Toxic alkaloid: Senecionine ‘responsible for bread-poisoning’ [62].
*S. inaequidens DC*	Toxic alkaloids (pyrrolizidine): Retrorsine, senecionine [42], senecivernine, integerrimine, and an analogue of retrorsine [63]. Possibly another chemotype with furanosesquiterpenes [58].
*S. inornatus* DC.	Toxic alkaloid (pyrrolizidine): O7-Angeloylretronecine [64]. Sesquiterpenes (furano): Furanoeremophilanes [58].
*S. isatideus* DC.	Vinyl-olefins: Polyunsaturated alkenes [65].
*S. latifolius* DC.	Toxic alkaloids: Retrorsine, isatidine, sceleratine, chlorodeoxysceleratine (merenskine), and the N-oxides of sceleratine and merenskine [66].
*S. leucanthemifolius* Poir.	Toxic alkaloids (pyrrolizidine): Integerrimine and senecionine [67]. Essential oils: α-hydroxy-p-cymene, carvacrol, nerol, carveol, and cis-α-bisabolene [68].
*S. linifolius* L.	Sesquiterpenes: furanoeremophilanes, i.e., maturinone and seven cacalohastin derivatives [69].
*S. lividus* L.	Sesquiterpenes: Eremophilanes [29]. Possibly essential oils: [70].
*S. longiscapus* Bojer ex DC.	Essential oils: Sabinene, elemicin, β-pinene, methyleugenol, α-pinene, and myrcene [71].
*S. lydenburgensis* Hutch. and Burtt Davy	Sesquiterpenes: Multiple cacalol derivatives [72].
*S. lyratus* Forssk.	Sterols: Sitosterol and stigmasterol. Triterpene: β-Amyrin [73].
*S. macrocephalus* DC.	Toxic alkaloids (pyrrolizidine): Traces of 7-senecioyl-9-sarracinylheliotridine and 7-angelyl-9-sarracinyl-heliotridine [74].
*S. macrospermus* DC.	Sesquiterpenes: Cacalohastin derivatives [75].
*S. madagascariensis* Poir.	Toxic alkaloids (pyrrolizidine): Senecivernine, senecionine, integerrimine, senkirkine, mucronatinine, retrorsine, usaramine, otosenine, acetylsenkirkine, desacetyldoronine, florosenine, and doronine [76].
*S. malacitanus* Huter	Toxic alkaloids (pyrrolizidine): Unnamed derivatives [77].
*S. mariettae* Muschl.	Toxic alkaloid (pyrrolizidine): Retrorsine [78].
*S. maritimus* L.f.	Alkanes: Polyunsaturated aklenes and -ynes. Furanosesquiterpene derivatives [79].
*S. massaicus* (Maire) Maire	Essential oil: *p*-Cymene, n-hexadecanoic acid, and docosane-11-decyl [80].
*S. mauricei* Hilliard and B.L.Burtt	Sesquiterpenes: Senmauricinol-(2-methyacrylat), hilliardinolisobutyrat, hilliardinol-(2-methylacrylat), 10β-hydroxy-1-oxo-6β-isobutyryloxy-2,3-dehydro-furanoeremophil-9-on, 2,3-desoxyhilliardinol-isobutyrat, 2,3-desoxohilliardinol-(2-methylacrylat), and 8,12-dioxo-7,11,9,10-tetradehydroeremophilan [81].
*S. microglossus* DC.	Sterols: Stigmasterol, sitosterol, dammaradienol, its 3-epimer, and the angelate. Sesquiterpenes: Germacrene D, y- and δ-cadinene, bisabolol, the angelate [41].
*S. oxyodontus* DC.	Sesquiterpene: Pentaynene sesquiterpene, bisabolene derivatives [82], and triquinane sesquiterpenes [83] *p*-Hydroxyacetophenone [82].
*S. oxyriifolius* DC.	Sesquiterpenes: Tricyclic bisabolol derivatives (epoxides) [84].
*S. paludaffinis* Hilliard	Sesquiterpenes: Cacalol and bisabolol derivatives [85].
*S. paniculatus* P.J.Bergius	Toxic alkaloids (pyrrolizidine): 7β-Angelyl-1-methylene-8α-pyrrolizidine, 7β-angelyl-1-methylene-8α-pyrrolizidine, 7β-angelyl-1-methylene8α-pyrrolizidine-4-oxide, 7-angelylhastanecine, 9-angelylhastanecine, 7-angelylplatynecine, 9-angelylplatynecine, 9-angelylplatynecine-4-oxide, sarracine, neosarracine, and retrorsine [86].
*S. paucicalyculatus* Klatt	Toxic alkaloids (pyrrolizidine): Retrorsine and isatidine [87].
*S. polyanthemoides* Sch.Bip.	Sesquiterpenes (furano): Cacalol derivatives [55]. Essential oils: Limonene, *p*-cymene, β-selinene, α-pinene, β-pinene, 1,8-cineole, caryophyllene oxide, and humulene epoxide II [88].
*S. pterophorus* DC.	Toxic alkaloids (pyrrolizidine): Retronecine, otonecine, platynecine, and rosmarinecine derivatives [89].
*S. pubigerus* L.	Sesquiterpenes: Germacrene D, bicyclogermacrene, beta-farnesene, and bisabolol derivatives. Toxic alkaloid (pyrrolizidine): acylpyrrole [90].
*S. purpureus* L.	Sesquiterpenes (eremophilenes): Diesters of seneremophilondiol, senescaposone and isosenescaposone, esterified with 4-methyl-5-acetoxy-pent-2-enoic acid [21].
*S. retrorsus* DC.	Toxic alkaloids (pyrrolizidine): Retrorsine, isatidine, and isatinecic acid [91].
*S. rhomboideus* Harv.	Sesquiterpenes: Eremophilene derivatives [21].
*S. roseiflorus* R.E.Fr.	Flavonoids: O-methylated, i.e., 5,4′-dihydroxy-7-dimethoxyflavanone [92].
*S. rosmarinifolius* L.f.	Toxic alkaloid (macrolide): Rosmarinine [62].
*S. ruwenzoriensis* S.Moore	Toxic alkaloids (pyrrolizidine): Isoline and bisline [93].
*S. sandersonii* Harv.	Sesquiterpenes: Diester and germacrene derivatives [94].
*S. schweinfurthii* O.Hoffm.	Toxic alkaloid (pyrrolizidine): 7β-Hydroxy-1-methylene-8α-pyrrolizidine *N*-oxide [95].
*S. serratuloides* DC.	Sterols: Phytosteroids and estran-3-one, 17-(acetyloxy)-2-methyl-, (2à,5à,17á) [96]. Used as a medicine to treat gynaecological and obstetric disorders [46].
*S. speciosus* Willd.	Toxic alkaloids (pyrrolizidine): 7-Senecioyl-9-sarracinylheliotridine and 7-angelyl-9-sarracinyl-heliotridine [74].
*S. squalidus* L.	Essential oils: *p*-Cymene and α-phellandrene [97].
*S. subrubriflorus* O.Hoffm.	Diterpenes: Sandaracopimarene derivatives. Sesquiterpene: Bisabolene derivative (4,7-oxide) [98].
*S. sylvaticus* L.	Sesquiterpenes: Alkenes, bicyclic derivatives, and furanosesquiterpenes [79].
*S. syringifolius* O.Hoffm.	Toxic alkaloids (pyrrolizidine): Angularine, rosmarinine, and 12-O-acetylrosmarine, together with their N-oxides [99].
*S. tamoides* DC.	Flavonoids: di-C-rhamnosylapigenin, mangiferin, and isomangiferin [100].
*S. umbellatus* L.	Sesquiterpenes: Furanoeremophilane derivatives [101].
*S. vernalis* Walds. and Kit.	Sesquiterpenes (furano): Cacalol derivatives [55].
*S. vira-vira* Hieron.	Toxic alkaloids (pyrrolizidine): Anacrotine, neoplatyphylline, uspallatine. Flavonoids: Quercitrin, rutin, isorhamnetin 3-O-β-robinobioside. Sterols: Sitosterol, campesterol, stimasterol, stimasta-3,5-dien-7-one, and stimasta-4,6-dien-3-one. Triterpenes: α-/β-amyrins [102].
*S. vulgaris* L.	Toxic alkaloids (pyrrolizidine): Senecionine, senecionine N-oxide, integerrimine N-oxide, seneciphylline N-oxide, retrorsine N-oxide, and spartioidine N-oxide [103]. Essential oils: α-Humulene, (*E*)-β-caryophyllene, terpinolene, ar-curcumene, and geranyl linalool [104].

## Data Availability

Not applicable.

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
