# Peer review of "Comment on Pyrrolizidine Alkaloids and Terpenes from *Senecio* (Asteraceae): Chemistry and Research Gaps in Africa"

_molecules, 2022, doi:10.3390/molecules27248868_

Round 1
Reviewer 1 Report
I think it is a good report, however the species with phytochemical studies should be investigated in more depth. For example, S. vulgaris, according to web of science there are 678 reports, some of them phytochemical studies Brown MS (1996), Flade J (2019), Xie WD (2010), among others.
There are several papers that were not considered, including studies of the genera Othonna and Senecio. Some species were reduced to synonymy, while others were included in another genus. (Villiers AJ 1999, South African Journal of Botany 65 (1), pp 110-112); also, Manning J (New synonyms and a new name in Asteraceae: Senecioneae from the southern African winter rainfall region. Bothalia, 40(1), 37-46.); Calvo J (Taxonomic Revision of the Eurasian/Northwestern African Senecio doria Group (Compositae) Systematic Botany, Volume 40, Number 3, October 2015, pp. 900-913(14), Beentje H. Three new species and some nomenclatural changes in Senecio (Compositae/Asteraceae: Senecioneae) in the Flora Zambesiaca área, New Bulletin (2019) 74:67.
Similarly, with respect to Table Nº2, there are a number of works that were not considered. Their authors are the following and species:
Mohamedt, AEH et al. (Senecio aegyptus var. Discoidesus)
Habib, AAM (Senecio aegyptius and S. desfontainei)
Liddle, JR (S. chrysocoma)
Thieme G (S. angustifolius)
Urones JG (S. gallicus)
Mohammadhosseine (S. gallicus)
Ramadan T (S. glaucus)
Ahmed S (S. hadiensis)
Bicchii C (S. inaequidens)
Wiedenfeld H. (S. inornatus)
Bredenkamp MW (S. latifolius)
Idrisii FEJ. (S.leucanthemifolius)
From the same Table Nº 81 species are mentioned, but there are 80.
Author Response
Dear reviewer, thank you for your detailed analysis of the current communication article. At our affiliation we are eager to find species that require more chemical analysis, so the objective of the current manuscript is about taking the initial steps to identify species that we seek to work with.
I think it is a good report, however the species with phytochemical studies should be investigated in more depth. For example, S. vulgaris ...
I appreciate that you spotted this one, as it is strange that I missed it. It is now corrected in Table 1 and added to Table 2.
There are several papers that were not considered, ... Some species were reduced to synonymy, while others were included in another genus.
I went through all of the papers that you highlighted regarding changes to the synonymity. Fortunately, the kew site, plants of the world online, was quite comprehensive and up to date, with the exception of one of those papers. I made the corrections to the manuscript, just to get the numbers of species up to date with the latest information available.
Similarly, with respect to Table Nº2, there are a number of works that were not considered. Their authors are the following and species:
I followed all of the leads you provided and amended Table 2 with any newer details. I also separated some species from subspecies, as their chemical character differed, making the separation relevant.
From the same Table Nº 81 species are mentioned, but there are 80.
After adding S. vulgaris to the table, and updating the details with two subspecies, the number is now 83.
Thank you for your time and effort.
Reviewer 2 Report
Manuscript title: Comment on Pyrrolizidine Alkaloids and Terpenes from Senecio (Asteraceae): Chemistry, Health and Research Gaps in Africa
This study has certain significance in the plant science community. However, revisions are necessary for the current version of the manuscript. The following questions to be addressed/considered may be helpful to improve the manuscript.
Major comments
· Insufficient Abstract: In the abstract, the main aim and background of the manuscript are missing, the current version it only highlights the result. In addition, it would be even better to have a sentence as a future perspective.
· The unit/abbreviation is not mentioned before, consider defining the abbreviation when mentioned for the first time…. Please check throughout the manuscript to define the abbreviations.
· Line 71-75, the aim or hypothesis of the study is clear, however, the approach is missing ….
· Lake of scientific literature to support the statements and findings throughout the manuscript…... I have made some suggestions for that and more need it….
· More information is needed for ALL TABLE captions and define the abbreviation and units that are used. And adjust the significant figures for the table and manuscript.
· I am not sure whether the ‘’Health Gaps’’ term from the title is well discussed neither in the abstract nor the manuscript. Please consider discussing it or rephrasing it.
Detailed comments:
Introduction:
Line 36: Please add ……. By lateral transfer of toxic alkaloids in tea plantations, and water contamination….
And then add the following references:
https://doi.org/10.1038/s41598-020-76586-1
https://doi.org/10.1016/j.scitotenv.2020.142822
https://doi.org/10.1021/acs.est.0c06411
Line 41-42: A complicated sentence, please revise and check the grammar
Line 54: There are even risk assessments from European Food Safety Authority on pyrrolizidine alkaloids in honeys, please check below:
https://doi.org/10.2903/j.efsa.2017.4908
Line 73-75: A complicated sentence, please revise and check the grammar
Phytochemistry of African Senecio section
Line 89-94: A reference is needed here.
Line 93: the font changed, please consider harmonizing the text…..
Conclusion
Important conclusions! However, the future perspectives for the following research are highly crucial here …..
Author Response
Dear Reviewer,
Thank you very much for taking the time to review the manuscript. It has improved so much with your help so far.
Major comments
- Insufficient Abstract: In the abstract, the main aim and background of the manuscript are missing, the current version it only highlights the result. In addition, it would be even better to have a sentence as a future perspective.
Revised as requested.
- The unit/abbreviation is not mentioned before, consider defining the abbreviation when mentioned for the first time…. Please check throughout the manuscript to define the abbreviations.
Yes a valuable point. No units are used in the manuscript, and only one abbreviation. I have defined this abbreviation as requested.
- Line 71-75, the aim or hypothesis of the study is clear, however, the approach is missing ….
I have revised to make the approach more obvious.
- Lake of scientific literature to support the statements and findings throughout the manuscript…... I have made some suggestions for that and more need it….
I have added many more references to Table 2, so that now it has lots of references.
- More information is needed for ALL TABLE captions and define the abbreviation and units that are used. And adjust the significant figures for the table and manuscript.
Yes, a very good point. I have not used any abbreviations in the tables, or numbers or units, but I did make significant edits to the tables, by adding more information to them.
- I am not sure whether the ‘’Health Gaps’’ term from the title is well discussed neither in the abstract nor the manuscript. Please consider discussing it or rephrasing it.
I have removed 'health' from the title.
Detailed comments:
Introduction:
Line 36: Please add ……. By lateral transfer of toxic alkaloids in tea plantations, and water contamination….
Completed as requested.
And then add the following references:
Completed as requested.
Line 41-42: A complicated sentence, please revise and check the grammar
Completed as requested.
Line 54: There are even risk assessments from European Food Safety Authority on pyrrolizidine alkaloids in honeys, please check below:
https://doi.org/10.2903/j.efsa.2017.4908
This is interesting, I have added a sentence about this to the introduction.
Line 73-75: A complicated sentence, please revise and check the grammar
Completed as requested.
Phytochemistry of African Senecio section
Line 89-94: A reference is needed here.
Completed as requested.
Line 93: the font changed, please consider harmonizing the text…..
Completed as requested.
Conclusion
Important conclusions! However, the future perspectives for the following research are highly crucial here …..
I have added another sentence to the conclusion that is the salient point I extracted from the manuscript.
Round 2
Reviewer 2 Report
The revised manuscript has improved compared to the original version. The authors tried to address my questions as much as possible. I recommend the manuscript to be published!